# A Pd/MnO₂ Electrocatalyst for Nitrogen Reduction to Ammonia under Ambient Conditions

**Chang Sun** **, Yingxin Mu and Yuxin Wang ***

State Key Laboratory of Chemical Engineering, School of Chemical Engineering, Tianjin University,
Tianjin 300350, China; sunchangtju@163.com (C.S.); muyingxin@tju.edu.cn (Y.M.)

\* Correspondence: yxwang@tju.edu.cn; Tel.: +86-22-2789-0515

**Abstract:** Electrochemical ammonia synthesis, which is an alternative approach to the Haber–Bosch process, has attracted the attention of researchers because of its advantages including mild working conditions, environmental protection, and simple process. However, the biggest problem in this field is the lack of high-performance catalysts. Here, we report high-efficiency electroreduction of $N_2$ to $NH_3$ on $\gamma$-MnO₂-supported Pd nanoparticles (Pd/$\gamma$-MnO₂) under ambient conditions, which exhibits excellent catalytic activity with an $NH_3$ yield rate of 19.72 $\mu g \cdot mg^{-1}_{Pd} \, h^{-1}$ and a Faradaic efficiency of 8.4% at −0.05 V vs. the reversible hydrogen electrode (RHE). X-ray diffraction (XRD) and transmission electron microscopy (TEM) characterization shows that Pd nanoparticles are homogeneously dispersed on the $\gamma$-MnO₂. Pd/$\gamma$-MnO₂ outperforms other catalysts including Pd/C and $\gamma$-MnO₂ because of its synergistic catalytic effect between Pd and Mn.

**Keywords:** electrochemical synthesis of ammonia; palladium; ambient conditions; MnO₂; synergistic catalytic

## 1. Introduction

Nitrogen is the most abundant gas in the atmosphere. The product of nitrogen fixation, especially ammonia ($NH_3$), not only is vital to life but also plays a key role in many other fields, such as transportation, refrigerants, and fertilizer. More than 80% of ammonia is used to manufacture chemical fertilizer [1]. Besides, ammonia is not only an important inorganic chemical product but also a key fuel and energy storage material. The flammability range is narrow, so it is easier to store and transport than liquid hydrogen [2,3]. The main way of nitrogen fixation in the industry is the Haber–Bosch process, which converts $N_2$ and $H_2$ into ammonia on Fe-based catalysts. However, the Haber–Bosch process needs harsh conditions, and the required energy comes from coal and natural gas, leading to its energy consumption occupies the world's energy supply more than 1% every year. Moreover, this process produces more than 300 million tons of $CO_2$ each year [4–6]. So it is necessary to develop new methods of ammonia synthesis which can avoid these drawbacks [7–11].

One promising approach to the Haber−Bosch process is to use electrical energy to drive the ammonia synthesis reaction [12–14]. This approach of ammonia synthesis has numerous advantages including low working temperature, low working pressure, environmental protection, and simple process, especially as the emission of $CO_2$ in this process is much lower than that in the Haber–Bosch process [15]. So electrochemical synthesis of ammonia has attracted the attention of researchers in recent years. At present, many electrocatalysts have been proved that have effective catalytic performance. Precious metal catalysts are the earliest reported batches of electrochemical ammonia synthesis catalysts, including Pt, Ir, Pd, Au, etc. The $NH_3$ yield rate of most precious metal catalysts is between $9 \times 10^{-11}$ mol/(s·cm²) and $10^{-12}$ mol/(s·cm²) [16–22]. Lan et al. studied the electrochemical synthesis of ammonia with water and air using Pt/C as the catalyst [23,24]. The maximum $NH_3$ yield rate

can reach $3.50 \times 10^{-9}$ mol/(s·cm$^2$). However, the Faradaic efficiency of Ir and Pt catalysts are extremely low because they are effective hydrogen evolution reaction (HER) catalysts. Many non-precious metal catalysts also exhibit effective catalytic performance. Fe is the most widely used catalyst for nitrogen fixation in the industry. Recently, inspired by natural nitrogen fixation, more Mo catalysts have also been reported [25–27]. Besides, more transition metals, including Mn and Ti, have also been reported for electrocatalytic reduction of dinitrogen to NH$_3$ [28,29]. However, most of the electrocatalysts have low activity and Faradaic efficiency for NH$_3$ production. Therefore, the improvement of electrocatalysts is essential for the progress of electrochemical ammonia synthesis, which needs a better design of N$_2$ reduction reaction (NRR) electrocatalyst and the electrochemical system [17,24,30–38]. Liu et al. reported an effective biomimetic strategy to boost electrocatalytic N$_2$ fixation, which shows that the effective design of the electrochemical system is important for the progress of electrochemical ammonia synthesis [39].

There are two major steps in the process of electrocatalytic synthesis of ammonia: N$_2$ adsorption on electrocatalyst and hydrogenation of nitrogen molecules on the surface of the electrocatalyst. Mukundan et al. demonstrated this process by using origami-like Mo$_2$C as the catalyst of electrochemical ammonia synthesis [40]. However, almost no electrocatalyst performs well in both steps. Skulason [41] proposed through simulation calculations that when the adsorption energy of N atoms on the surface of electrocatalyst is bigger than that of H atoms, N atoms cover most of the surface of the catalyst to improve NRR selectivity while suppressing hydrogen evolution reaction (HER). Studies [42,43] have shown that Mn, Re, and other pre-transition metal elements have strong adsorption energy for N$_2$ molecules. But these elements are usually inefficient in the hydrogenation of N$_2$ molecules, which is another major step in the electrocatalytic synthesis of ammonia. However, Pd, Ni, and other post-transition metal elements perform well in hydrogenation reactions, but have low adsorption energy of N$_2$. Especially for Pd, it is easy to absorb hydrogen atoms in its lattice to form PdH, and can achieve hydrogenation reaction easier than other metal catalysts through the Grotthuss-like proton-hopping mechanism [21,44–46]. Consequently, combining multiple catalysts with different advantages in the ammonia synthesis process is a promising design for NRR electrocatalyst.

In this study, we effectively load Pd nanoparticles on the surface of Mn by the polyol reduction method and report high-performance electroreduction N$_2$ to NH$_3$ on Pd/γ-MnO$_2$ under ambient conditions, which exhibits high activity and selectivity with an NH$_3$ yield rate of 19.72 µg·mg$^{-1}$ $_{Pd}$ h$^{-1}$ ($6.44 \times 10^{-11}$ mol·s$^{-1}$ cm$^{-2}$) and a Faradaic efficiency of 8.4% at −0.05 V vs. the reversible hydrogen electrode (RHE). The result shows that γ-MnO$_2$ is the best carrier of Pd among the three crystal forms of MnO$_2$. The performance of Pd/γ-MnO$_2$ demonstrates the synergistic catalytic effect between Pd and Mn, which is an promising design strategy for NRR catalysts.

## 2. Results and Discussion

### 2.1. Characterization of Catalyst

The MnO$_2$ were prepared by the hydrothermal method and the preparation method of Pd/γ-MnO$_2$ and Pd/C catalysts was polyol reduction method. Scanning electron microscope (SEM) image and X-ray diffraction (XRD) pattern of γ-MnO$_2$ were shown in Figure 1. γ-MnO$_2$ has a flower-like structure, suggesting that it's a suitable carrier for metal particles. The peaks in Figure 1b with 2θ values of 22.1°, 37.1°, 42.2°, 55.8°, and 67.8° can be indexed to the diffraction from (101), (210), (211), (212), and (511) lattice planes of γ-MnO$_2$ (JCPDS #44-0142).

As shown in Figure 2a, the Pd nanoparticles are loaded on the surface of carbon black XC-72. Figure 2b is the transmission electron microscopy (TEM) images of γ-MnO$_2$ without Pd particles on it. Figure 3 shows TEM and energy dispersive x-ray spectroscopy (EDX) mapping images of the obtained Pd/γ-MnO$_2$ catalysts, which suggest that the Pd nanoparticles are more evenly dispersed on γ-MnO$_2$ than those on the carbon black. Figure 3a,b show that Pd nanoparticle average sizes around 4 nm. Figure 3b shows the atomic lattice fringes of the Pd particles with lattice plane spacings determined to

be 0.225 nm, corresponding to the (111) lattice spacing of Pd. Figure 3c,d were EDX mapping images of Pd and Mn elements on Pd/γ-MnO$_2$, demonstrate that Pd can be detected throughout the surface of γ-MnO$_2$.

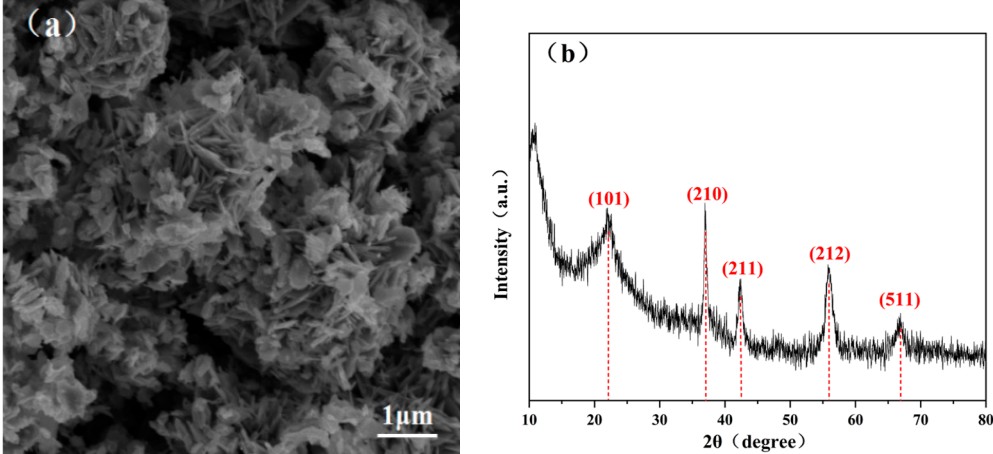

**Figure 1.** Scanning electron microscope (SEM) image (**a**) and X-ray diffraction (XRD) pattern (**b**) of γ-MnO$_2$.

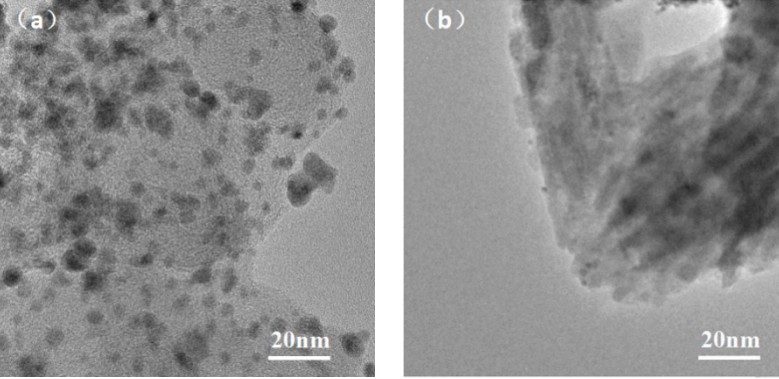

**Figure 2.** Transmission electron microscopy (TEM) images of (**a**) Pd/C and (**b**) γ-MnO$_2$.

Figure 4 shows the X-ray diffraction (XRD) pattern of the Pd/γ-MnO$_2$ catalyst. It was found that the diffraction peaks at 22.1°, 37.1°, 42.2°, and 55.8° correspond to (101), (210), (211), and (212) lattice planes of γ-MnO$_2$ (JCPDS #44-0142), and the peaks at 40.0°, 46.6°, and 68.1° correspond to (111), (200), and (220) lattice planes of Pd (JCPDS#65-2867). Figures 3 and 4 demonstrate that Pd nanoparticles were successfully dispersed on the surface of γ-MnO$_2$ by the polyol reduction method.

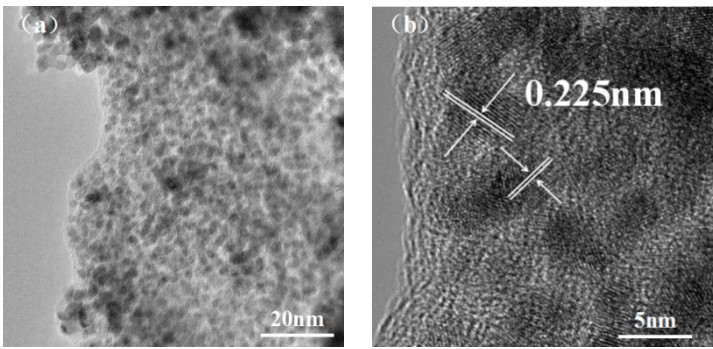

**Figure 3.** *Cont.*

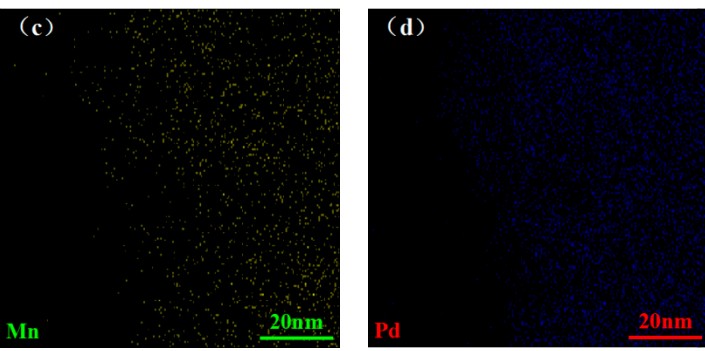

**Figure 3.** TEM images (**a**,**b**) of Pd/γ-MnO$_2$, and energy dispersive x-ray spectroscopy (EDX) mapping images (**c**,**d**) of Mn and Pd elements on Pd/γ-MnO$_2$.

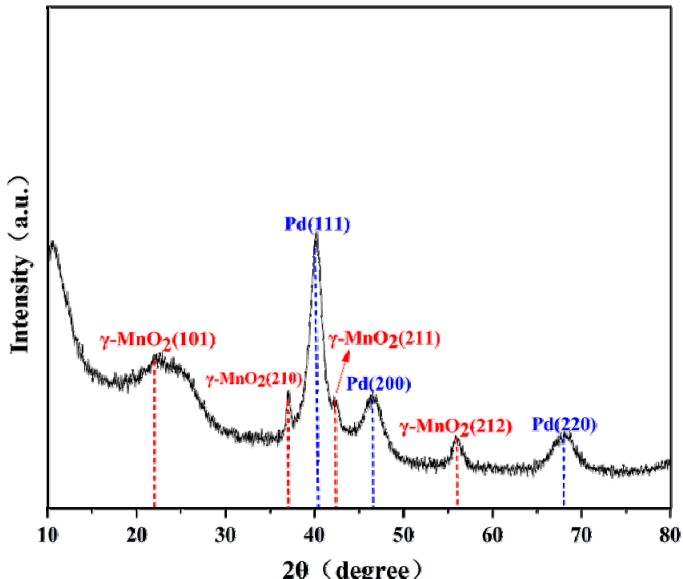

**Figure 4.** XRD pattern of Pd/γ-MnO$_2$ catalyst.

## 2.2. Electroreduction of N$_2$ to NH$_3$ on Pd/γ-MnO$_2$ Catalyst

The electrolysis experiments were performed using a gas-tight single compartment electrochemical cell which is filled with 0.1 M KOH as the electrolyte. A piece of Pt gauze and saturated calomel electrode were used as counter electrode and reference electrode, respectively. N$_2$ gas was delivered into the cell by N$_2$ gas bubbling. The NRR activities and Faradaic efficiencies of the electrodes were measured by controlled potential electrolysis with N$_2$-saturated electrolyte for 3 h. The NH$_3$ produced by electrode was quantified at the end of each electrolysis process using the calibration curves established by the indophenol blue method.

To illustrate whether Pd and Mn have a synergistic catalytic effect for electrochemical ammonia synthesis. We compared the catalytic performance of the Pd/γ-MnO$_2$ catalyst with Pd/C and γ-MnO$_2$ catalysts in three N$_2$-saturated electrolytes at −0.05 V vs. RHE. As is shown in Figure 5, the NH$_3$ yield rate and Faradaic efficiency on Pd/γ-MnO$_2$ catalyst are both higher than those on Pd/C and γ-MnO$_2$ catalysts. The NH$_3$ yield rate on Pd/γ-MnO$_2$ reaches 3.94 μg·mg$^{-1}_{cat}$ h$^{-1}$, which is around two times of that on Pd/C (2.11 μg·mg$^{-1}_{cat}$ h$^{-1}$) and around four times of that on γ-MnO$_2$ (1.17 μg·mg$^{-1}_{cat}$ h$^{-1}$). Considering the Pd loading is 20 wt.%, it can reach an NH$_3$ yield rate of 19.72 μg·mg$^{-1}_{Pd}$ h$^{-1}$ on Pd/γ-MnO$_2$.

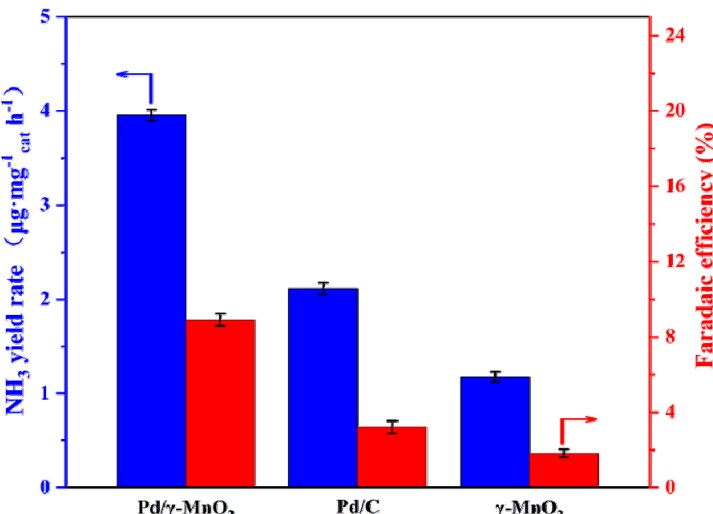

**Figure 5.** Comparison of the Pd/γ-MnO$_2$ catalyst with Pd/C and γ-MnO$_2$ catalysts for catalytic performance at -0.05 V in 0.1 M KOH.

Figure 6 shows that the NH$_3$ yield rates on Pd/γ-MnO$_2$ catalysts are invariably higher than those on the other two catalysts in a wide potential range, and reaches a maximum value at −0.05 V. The γ-MnO$_2$ catalyst shows a low NH$_3$ yield rate and Faradaic efficiency without Pd particles. However, γ-MnO$_2$-supported Pd nanoparticles outperform the Pd/C catalyst, indicating a synergistic catalytic effect for electroreduction of N$_2$ to NH$_3$ between Pd and Mn. The Pd/γ-MnO$_2$ catalysts combine different advantages of Pd and Mn. The reaction active site is at the phase interface of Pd and γ-MnO$_2$. To be specific, Mn adsorbs a large amount of N$_2$ at first because of its strong adsorption capacity for nitrogen molecules [29], and then Pd achieves hydrogenation for N$_2$ through its unique hydrogenation ability (Grotthuss-like proton-hopping mechanism) [21]. This synergy effect significantly accelerates the two major steps in the ammonia synthesis process. And Pd nanoparticles is evenly dispersed on the surface of γ-MnO$_2$, with a huge electrochemically active area, resulting in its unique catalytic activity. The Pd/γ-MnO$_2$ catalyst achieves an NH$_3$ yield rate and Faradaic efficiency that are comparable to the recently reported catalysts for NRR under ambient conditions. The less favorable kinetics of HER on Pd/γ-MnO$_2$ is because of its higher barrier for mass and charge transfer, as evidenced by the electrochemical impedance spectra (EIS) in Figure 7. The EIS image of Pd/γ-MnO$_2$ catalyst has a larger radius than Pd/C and γ-MnO$_2$, which indicates its larger barrier to HER. Therefore, the Pd/γ-MnO$_2$ catalyst has the highest NRR selectivity among the three.

In addition, we have investigated the NRR activity and Faradaic efficiency of Pd/α-MnO$_2$ and Pd/β-MnO$_2$ at −0.05 V. As shown in Figure 8, the Pd/γ-MnO$_2$ catalyst has superior NRR activity and selectivity than Pd/α-MnO$_2$ and Pd/β-MnO$_2$, indicating that γ-MnO$_2$ is the optimal carrier of Pd nanoparticles among these three crystal forms of MnO$_2$. This result is due to their different microstructures. As shown in Figure 9, the aggregation of Pd particles was easier to happen when we disperse Pd on α-MnO$_2$ and β-MnO$_2$. Pd particles can be more homogeneously dispersed on the surface of γ-MnO$_2$ than α-MnO$_2$ and β-MnO$_2$, and have a smaller particle size, which increases the catalyst surface area. The reactive sites of NRR possibly at the interface between the Pd and Mn. Therefore, the homogeneous dispersion of Pd nanoparticles provides more electrochemical reactive sites for NRR, while suppressing the competing HER. The aggregation of Pd particles on Pd/α-MnO$_2$ and Pd/β-MnO$_2$ leads to the more favorable kinetics of HER, as evidenced by the EIS image in Figure 10.

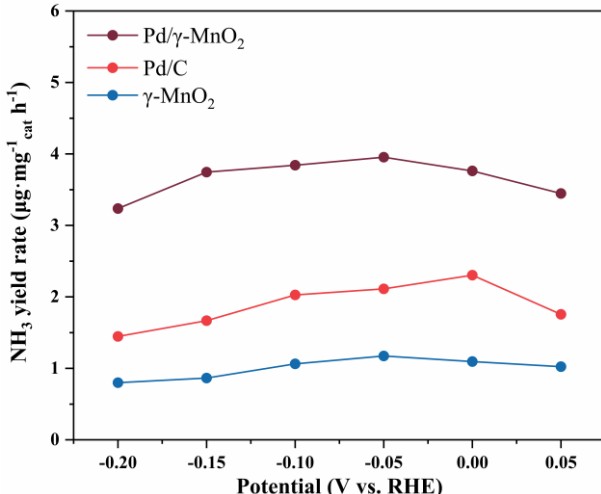

**Figure 6.** The NH$_3$ yield curves on Pd/γ-MnO$_2$, Pd/C, and γ-MnO$_2$ catalysts at various potentials.

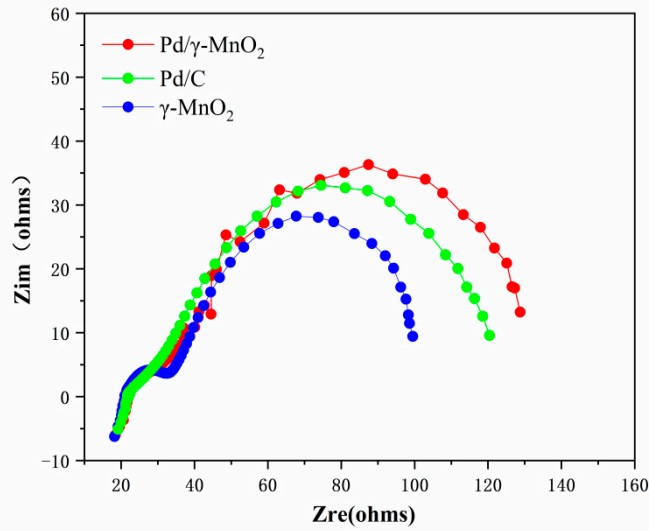

**Figure 7.** Electrochemical impedance spectra of Pd/γ-MnO$_2$, Pd/C, and γ-MnO$_2$ catalysts.

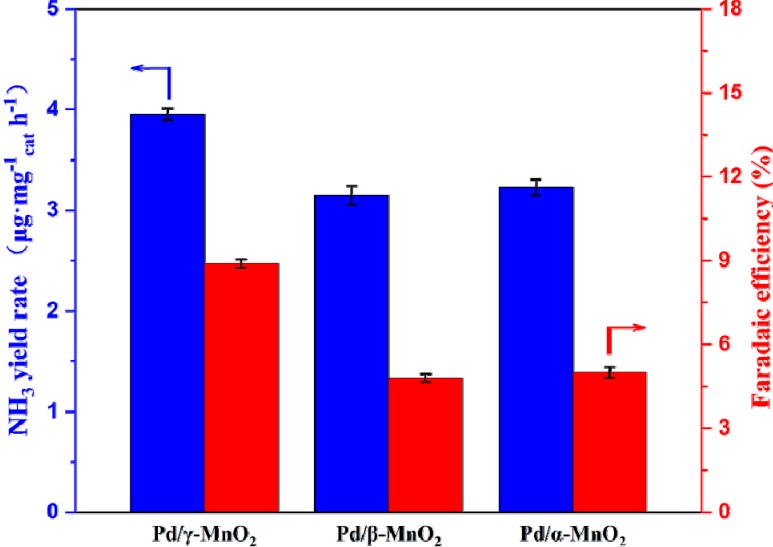

**Figure 8.** Comparison of the Pd/γ-MnO$_2$ catalyst with Pd/α-MnO$_2$ and Pd/β-MnO$_2$ catalysts for catalytic performance at −0.05 V in 0.1 M KOH.

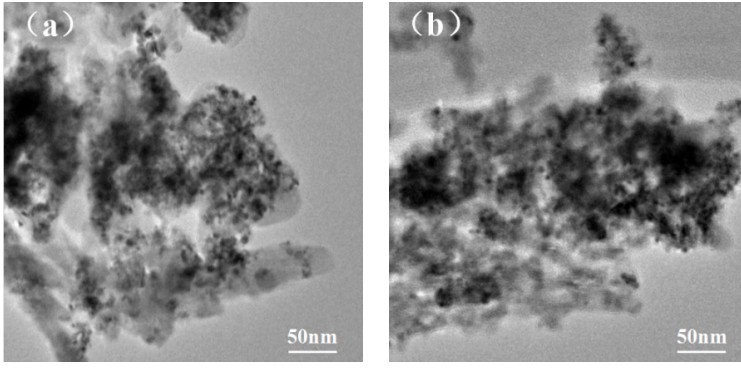

**Figure 9.** TEM images of (**a**) Pd/α-MnO₂ and (**b**) Pd/β-MnO₂ catalysts.

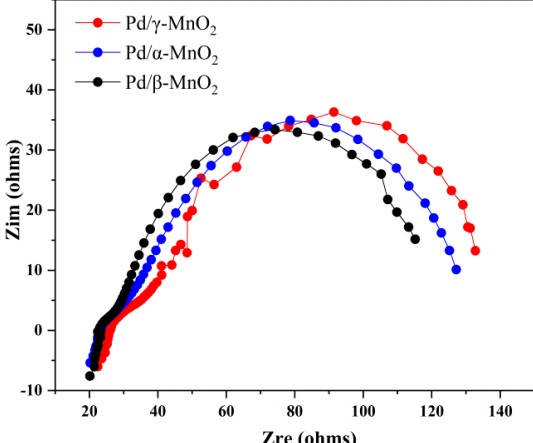

**Figure 10.** Electrochemical impedance spectra of Pd/α-MnO₂, Pd/β-MnO₂, and Pd/γ-MnO₂ catalysts.

In the preparation of Pd/γ-MnO₂ (see Materials and Methods for details of catalysts preparation), the pH value of the Pd precursor (K₂PdCl₄) solution will also affect the aggregation of Pd nanoparticles. When the pH value of the K₂PdCl₄ was 3, 5, 9, and 11, there is obvious aggregation on the γ-MnO₂, as shown in Figure 11. However, Pd nanoparticles can be homogeneously dispersed on γ-MnO₂ under the condition that the pH value of K₂PdCl₄ is 7. Consequently, the Pd/γ-MnO₂ prepared under the condition that pH = 7 can reach a higher NH₃ yield rate and Faradaic efficiency than that on other catalysts in Figure 12. It also suggests that the aggregation of Pd particles will cause a loss of performance. Figures 11 and 12 suggest that the conditions of strong acidity and alkalinity are unfavorable for Pd²⁺ reduction. This can be recognized as important inspiration in the design of catalysts of electrochemical synthesis of ammonia. The conditions in the catalyst preparation process, such as heating temperature and stirring time, may have a great influence on the catalytic performance and microstructure of the catalysts.

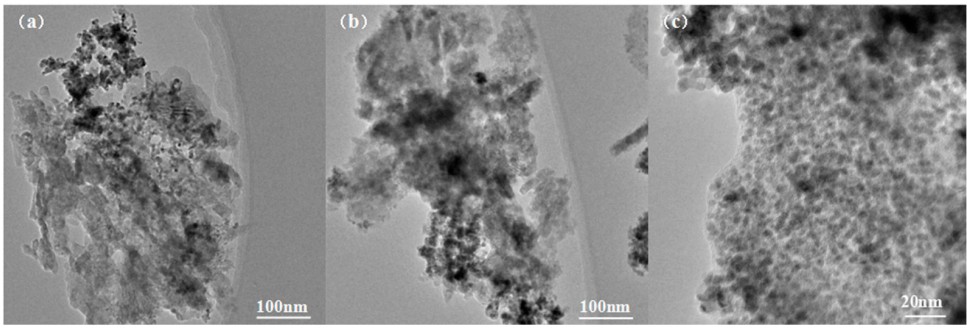

**Figure 11.** *Cont.*

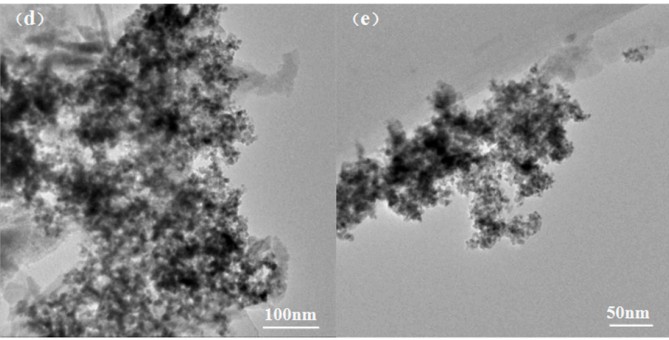

**Figure 11.** TEM images of Pd/γ-MnO$_2$ catalysts prepared under different pH value. (**a**) pH = 3; (**b**) pH = 5; (**c**) pH = 7; (**d**) pH = 9; and (**e**) pH = 11.

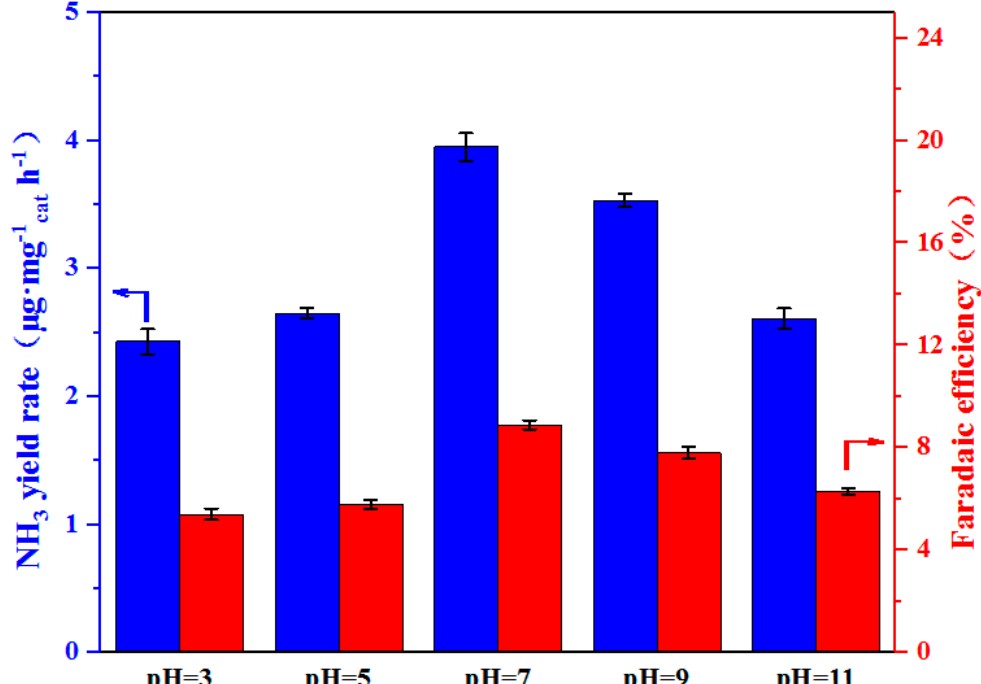

**Figure 12.** Ammonia synthesis rates and Faradaic efficiencies of Pd/γ-MnO$_2$ prepared under different pH conditions.

Moreover, we have tested the N source of the produced ammonia. First, we performed control experiments with Ar-saturated electrolyte or without Pd/γ-MnO$_2$ catalyst. As shown in Figure 13, no apparent NH$_3$ was detected when the bubbled N$_2$ gas was replaced by Ar or when a carbon paper electrode without the Pd/γ-MnO$_2$ catalyst was used. And the 0.1 M KOH without electrodes cannot detect ammonia as well. It suggests that the NH$_3$ was produced by N$_2$ reduction in the presence of Pd/γ-MnO$_2$ catalyst.

The stability of the Pd/γ-MnO$_2$ catalyst for electroreduction of N$_2$ to NH$_3$ was evaluated by consecutive recycling electrolysis. As shown in Figure 14a, only a slight decline in the total current was observed. However, Figure 14b shows that the NH$_3$ yield rate and Faradaic efficiency decreased to 2.45 μg·mg$^{-1}$$_{cat}$ h$^{-1}$ (12.25 μg·mg$^{-1}$$_{Pd}$ h$^{-1}$) and 5.2%, indicating a loss of electrochemical activity by 40% after the 12 h operation. This loss in the electrochemical activity and selectivity is due to the loss of active Pd surface area caused by the aggregation of Pd nanoparticles on the surface of Pd/γ-MnO$_2$, as evidenced by TEM images of the Pd/γ-MnO$_2$ catalyst before and after stability test (Figure 15).

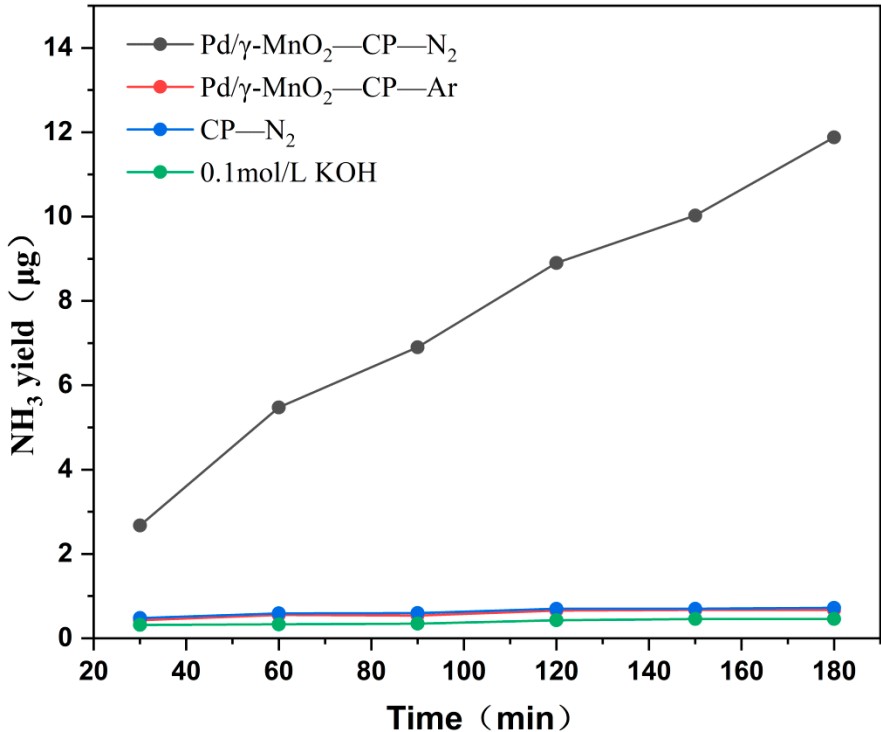

**Figure 13.** The curve of ammonia yield and time under four different conditions.

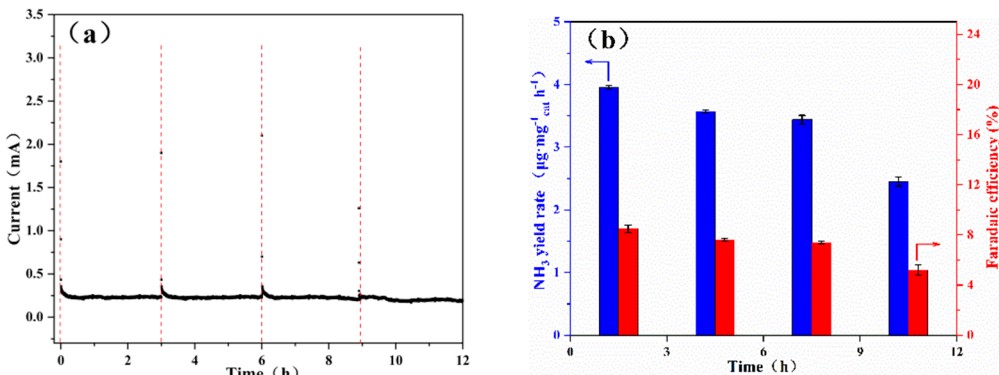

**Figure 14.** (**a**) Variation of electric current in the four successive operations each lasting 3 h. (**b**) Ammonia synthesis rates and Faradaic efficiency of each electrolysis cycle.

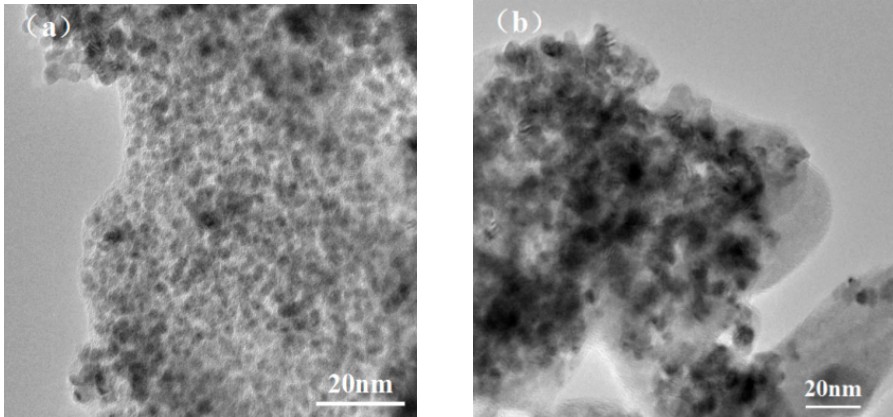

**Figure 15.** TEM images of the Pd/γ-MnO₂ catalyst before (**a**) and after (**b**) stability test.

## 3. Materials and Methods

### 3.1. Materials

Potassium hydroxide and ethylene glycol were obtained from Tianjin Kemiou Chemical Reagent Company (Tianjin, China). Salicylic acid and ammonium sulfate were supplied by Tianjin Seans Biochemical Technology Company (Tianjin, China). $N_2$ and Ar were supplied by Tianjin Dongxiang Special Gas Company (Tianjin, China). Palladium dichloride was obtained from Ailan (Shanghai) Chemical Technology (Shanghai, China). Sodium borohydride, isopropanol and concentrated sulfuric acid were obtained from Tianjin Jiangtian Chemical Technology (Tianjin, China). Sodium hydroxide was supplied by Tianjin Guangfu Technology Development Company (Tianjin, China).

### 3.2. Catalyst Preparation

#### 3.2.1. $\alpha$-$MnO_2$

First, 1 g of $KMnO_4$ and 0.5 g of $MnCl_2 \cdot 4H_2O$ were dissolved in 140 mL of water, followed by sonication for 30 min. Then the solution was poured into a 200 mL PTFE-lined hydrothermal reactor and reacted for 12 h at 180 °C. The mixture was filtered with suction and rinsed with deionized water after the reaction. The resulting $\alpha$-$MnO_2$ was dried at 60 °C overnight.

#### 3.2.2. $\beta$-$MnO_2$

First, 0.632 g of $KMnO_4$ was dissolved in 112 mL of water, and 11 mL of concentrated hydrochloric acid (12 mol/L) was added dropwise with stirring, followed by sonication for 3 h. Then the solution was poured into a 200 mL PTFE-lined hydrothermal reactor and reacted for 12 h at 160 °C. The mixture was filtered with suction and rinsed with deionized water after the reaction. The resulting $\beta$-$MnO_2$ was dried at 60 °C overnight.

#### 3.2.3. $\gamma$-$MnO_2$

0.92 g $(NH_4)S_2O_8$ and 0.4 g $MnCl_2 \cdot 4H_2O$ were dissolved in 60 mL of water respectively and recorded as solution A and solution B. Next, solution B was added dropwise to solution A with stirring. Then the solution was poured into a 200 mL PTFE-lined hydrothermal reactor and reacted for 24 h at 90 °C. The mixture was filtered with suction and rinsed with deionized water after the reaction. The resulting $\gamma$-$MnO_2$ was dried at 60 °C overnight.

#### 3.2.4. Pd/$\gamma$-$MnO_2$

We use polyol reduction method to prepare the Pd/$\gamma$-$MnO_2$ catalyst. First, $K_2PdCl_4$ (the Pd precursor solution) was prepared by dissolving palladium dichloride in water with KCl. And 120 mg of $\gamma$-$MnO_2$ was dispersed in 120 mL of ethylene glycol, followed by sonication for 1 h. Then, 5 mL of $K_2PdCl_4$ solution (containing Pd 6 mg/mL) was added into this mixture. After stirring for 30 min at room temperature, the mixture was heated at 130 °C for 2 h. The mixture was filtered with suction and rinsed with deionized water. Then we dried the product at 60 °C for 12 h, with a Pd loading of 20 wt.%. Pd/C, Pd/$\alpha$-$MnO_2$, and Pd/$\beta$-$MnO_2$ catalysts with a Pd loading of 20 wt.% were synthesized by the same procedure, except with different carriers.

### 3.3. Preparations of the Working Electrodes

First, 2 mg of $\gamma$-$MnO_2$-supported Pd nanoparticles catalyst was dispersed in diluted Nafion alcohol solution containing 1 mL ethanol and 45 μL Nafion, which formed a homogeneous suspension after sonication for 2 h. The carbon-paper electrodes were prepared by drop-casting the suspension on carbon paper ($1.25 \times 0.8$ cm$^2$), with a total mass loading of 1 mg (of which 20 wt.% is Pd).

### 3.4. Ammonia Quantification

In this study, $NH_3$ was quantitatively determined by indophenol blue method. Each sample solution was added with 0.5 mL of 5% salicylic acid solution, 0.1 mL of 1% sodium nitroprusside solution, and 0.1 mL of 0.05 mol/L NaClO solution. Leave at room temperature for 1 h in the dark. Then measure its absorbance by ultraviolet spectrophotometer, contrast with the calibration curve.

### 3.5. Calculation of the $NH_3$ Yield Rate the Faradaic Efficiency

$NH_3$ yield rate the Faradaic efficiency were calculated as follows:

$$\text{Faradaic efficiency} = (3F \times c_{NH_3} \times V)/Q$$

$$\text{Yield rate} = (17c_{NH_3} \times V/(t \times m)$$

F is the Faraday constant (96485 C·mol$^{-1}$); $c_{NH_3}$ is the concentration of $NH_3$; V is the volume of the electrolyte; Q is the charge flowed through the electrode; t is the reaction time; and m is the mass of the whole catalyst or precious metal.

## 4. Conclusions

In this article, we use polyol reduction method to prepare the $\gamma$-$MnO_2$-supported Pd nanoparticles catalyst (Pd/$\gamma$-$MnO_2$), which exhibits effective catalytic activity for the electrochemical ammonia synthesis. The $NH_3$ yield rate and Faradaic efficiency of Pd/$\gamma$-$MnO_2$ catalyst reaches a maximum value of 19.72 µg·mg$^{-1}$$_{Pd}$ h$^{-1}$ ($6.44 \times 10^{-11}$ mol·s$^{-1}$ cm$^{-2}$) and 8.4%, respectively, at −0.05 vs. RHE, indicating a synergistic catalytic effect between Pd and Mn. Pd/$\gamma$-$MnO_2$ outperforms other catalysts including Pd/C and $\gamma$-$MnO_2$ because of its synergistic catalytic effect. Moreover, our result shows that $\gamma$-$MnO_2$ is the optimal carrier for Pd nanoparticles among three different crystal forms of $MnO_2$. Further improvement in research of the interaction between Pd and Mn will be beneficial for the NRR activity and selectivity of the catalyst.

**Author Contributions:** Conceptualization, methodology, Y.W. and C.S.; formal analysis, validation, and review, Y.W., C.S., and supervision, project management, Y.W., Y.M.; Experimental work and draft writing, C.S. All authors have read and agreed to the published version of the manuscript.

**Funding:** This research received no external funding.

**Acknowledgments:** This work has been supervised and reviewed by Luofu Min and Lu Liu.

**Conflicts of Interest:** The authors declare no conflict of interest.

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
