# Peer review of "A Pd/MnO2 Electrocatalyst for Nitrogen Reduction to Ammonia under Ambient Conditions"

_catalysts, doi:10.3390/catal10070802_

Round 1

Reviewer 1 Report

Dear Authors,

I have found your paper as well written and well presented.

In figure 12 there are a lack of error bars. Please add them.

I would suggest to make English language revision and the article is ready to be published

Reviewer 2 Report

The authors prepared Pd supported on g-MnO2 and study their nitrogen reduction reaction catalytic activity to produce ammonia from nitrogen. The rates are comparable to some of the recent literature reports. The authors also have followed some of the protocols set by leading experts in this field to carry out control measurements. However, there are important concerns that need to be addressed.

  1. The units for yield rate should be mols /s /cm2. While many recent reports are using the units reported in this manuscript, the preferred and correct unit is mols /s /cm2. A conversion into conventional units indicate a rate of 6x10-11 mols /s /cm2 which is about four orders of magnitude lower then the commercialization requirements.
  2. Introduction could be a bit more elaborate discussing other potential electrocatalysts or the rates required to achieve commercial viability for the EC-NRR process.
  3. All the reported XRD patterns need JCPDS numbers for comparison.
  4. Figure 2 and 3, the 20 nm scale picture shows the presence of flakes for g-MnO2 in Figure 2b but when deposited with Pd nanoparticles as shown in Figure 3, the flakes seem to be disappeared. Further the catalyst density in Pd/g-MnO2 seems to be higher than PdC. What is the metal/support ratio in these catalyst systems?
  5. Figure 5, it looks like all the tested catalyst systems show NRR catalytic activity and inclusion of Pd on g-MnO2 merely increases its catalytic activity. In this instance, what is the origin of NRR activity on g-MnO2 is not clear…
  6. Between Figure 5 and 6, the same catalyst system shows a yield rate of ~20 and 4 micrograms of ammonia. why is this discrepancy?
  7. Line 138 – 140, the authors claim that Mn adsorbs N2 better and incorporation of Pd helps the hydrogenation of these adsorbed N2 due to Pd’s hydrogenation ability. Both these claims need to be supported with citations. Further, the concept of one surface adsorbing nitrogen while the second surface adsorb and transfer hydrogen facilitating the NRR is recently published by Mukundan et al in Journal of the Electrochemical Society, 167, 044506”. This needs to be cited.
  8. Figure 7, Considering the higher hydrogenation properties of Pd and its known HER activity, why Pd based catalysts show a poor kinetics toward HER than g-MnO2 is not clear (lines 143 – 145).
  9. The comparison of Pd supported on alpha, beta and gamma MnO2 followed by effect of pH on Pd catalyst preparation is very contradicting to each other. For example, the authors claim gamma MnO2 provides a more evenly distributed Pd nanoparticles and is the reason for its improved NRR activity between the three supports. However, when upon varying the pH, the Pd agglomerates even on gamma MnO2. When comparing the aggregated Pd on alpha, beta and gamma MnO2 surfaces (Figure 8 and 12), there is no difference in catalytic activity. Is the improved performance solely due to a catalyst particle size effect?
  10. On both EIS data presented, why is there a variation in high frequency intercept if the electrolyte is the same in all experiments…
  11. A figure displaying the UV-vis spectra obtained for the indophenol blue samples and the calibration data set should be provided.
  12. Figure 15, what is (a) and (b) needs to be mentioned in the caption.

Reviewer 3 Report

Catalysts

Manuscript ID: 859668 

Title: A Pd/MnO2 Electrocatalyst for Nitrogen Reduction to Ammonia under Ambient Conditions.

Reviewer’s comments

In the present manuscript authors prepared γ-MnO2-supported Pd nanoparticles (Pd/γ-MnO2) for electroreduction N2 to NH3 under ambient conditions. The experimental results indicated that the prepared nanoparticles catalyst Pd/γ-MnO2 exhibited high activity and selectivity for the electrochemical synthesis of ammonia.

In my opinion, this a good work where the authors performed interesting physicochemical and electrochemical characterizations. However they did not explore enough and in depth the catalytic mechanisms of nitrogen hydrogenation and the relation between electrochemical parameters with the catalytic ones. For this reason, more experiments are necessary to be reported. As such, unfortunately, I cannot recommend this work for publication to Catalysts as is. Additionally, the following revisions should be taken into consideration if resubmitted:

  • The novelty should be better pointed out, taking into account works appeared the period 2015-2020 (and appropriately enhance the introduction section).
  • The manuscript should be enhanced with recently published works, especially of 2018-2020. (Nature Catalysis, 2018, 490–500 & Applied Catalysis B: Environmental, 2020, 118919 etc).
  • Moreover, the contribution should also be clearly pointed out. The authors should list in a table the more important results appeared the last decade in Intl literature and compare them with the results obtained here pointing out the relevance of the present work.
  • In the manuscript authors reported that “the Pd/γ-MnO2 catalyst achieves an NH3 yield rate and Faradaic efficiency that are comparable to the recently reported catalysts for NRR under ambient conditions” can they give us the reference? With which catalyst the comparison is made with?
  • The manuscript lacks information. Authors should perform linear sweep voltammetry (LSV) tests in an Ar and N2 saturated environment to qualitatively distinguish between HER and NRR. The current density is a crucial factor for the evaluation of the catalyst.
  • Figure 3 shows STEM or EDX mapping images? Please check and clarify it in the text.
  • As can be seen from figures 14 the stability of the tested materials is not good even for small periods of time. Could the authors commented this fact?
  • Looking at figure 5 I am thinking about the main mechanism of NRR is. Could the authors shed light in this? The nature and the role of the active sites (in each case) should be better and clearly pointed out.
  • English needs slight improvement.

Round 2

Reviewer 2 Report

I think the reviewers addressed most of my concerns. I am not satisfied by the response to the point 9. It still looks to be a catalytic activity improvement by improving particle distribution and size reduction. Also the response to point 11, the  authors has provided the calibration data but has not provided the actual UV-Vis spectra. The rest of the manuscript looks fine.

Reviewer 3 Report

The authors took into consideration all the suggestions I did and appropriately improved their manuscript.

I think now the paper it could be accepted for publications to Catalysts

Author Response

Dear Editors and Reviewers:

Thank you for your approval of our manuscript entitled “A Pd/MnO2 Electrocatalyst for Nitrogen Reduction to Ammonia under Ambient Conditions”(catalysts-859668). We appreciate for your warm work earnestly.

Once again, thank you very much for your comments and suggestions!

With best regards,

Chang Sun